# Heart Diseases Diagnose via Mobile Application

Mesut Güven [1,*] , Fırat Hardalaç [1], Kanat Özışık [2] and Funda Tuna [2]

1   Electrical and Electronics Engineering Department, Gazi University, Ankara 06570, Turkey; firat@gazi.edu.tr
2   Cardiovascular Surgery Department, Ankara City Hospital, Ankara 06800, Turkey;
    sozisik2002@yahoo.com (K.Ö.); ftuna02@yahoo.com (F.T.)
*   Correspondence: mesut.guven@gazi.edu.tr

**Featured Application: The design and verification result of a mobile application that can detect heart abnormalities is presented within the scope of this work.**

**Abstract:** One of the oldest and most common methods of diagnosing heart abnormalities is auscultation. Even for experienced medical doctors, it is not an easy task to detect abnormal patterns in the heart sounds. Most digital stethoscopes are now capable of recording and transferring heart sounds. Moreover, it is proven that auscultation records can be classified as healthy or unhealthy via artificial intelligence techniques. In this work, an artificial intelligence-powered mobile application that works in a connectionless fashion is presented. According to the clinical experiments, the mobile application can detect heart abnormalities with approximately 92% accuracy, which is comparable to if not better than humans since only a small number of well-trained cardiologists can analyze auscultation records better than artificial intelligence. Using the diagnostic ability of artificial intelligence in a mobile application would change the classical way of auscultation for heart disease diagnosis.

**Keywords:** heart diseases; auscultation; machine learning; telemedicine; digital stethoscope





## 1. Introduction

Heart disease is one of the most common causes of death worldwide [1]. Even in developed countries, healthcare services are expensive, and having a check-up in a clinic is not only a time-consuming and costly task, but it is also a relatively dangerous endeavor under the current conditions of a global pandemic. Therefore, telemedicine services and applications have become popular and gained much importance in recent years [2,3]. There is a consensus that telemonitoring capabilities which include the home assessment of various health parameters such as blood pressure, heart condition, weight, etc. should be enhanced and put at the disposal of patients [4]. Automatic diagnoses of heart diseases have been studied for a long time and the proposed algorithms have acceptable success on the largest available dataset, the PhysioNet/CinC Challenge 2016 dataset [5,6].

The general methodology of the automated heart sound classification algorithms consists of three steps. Those are segmentation, feature extraction, and classification steps (Figure 1). Segmentation is necessary for classification since most of the features used for classification are derived from fundamental heart sounds (FHSs) that occur because of the contraction and relaxation movements of the heart [7]. Even though there are some other lower-pitched sounds, FHSs usually include two distinct sounds, first (S1) and second (S2) heart sounds. The time-lapse between S1 and S2 is known as the diastolic period and the time lapse between S2 and S1 is known as the systolic period.

After successfully segmenting the heart sounds, features are extracted from the time, frequency, energy, and high order statistics domain. Then, a suitable classification algorithm is determined. Various methods have been employed for classification such as k-nearest neighbors, artificial neural networks, support vector machines, and hidden Markov models [8–10]. It is possible to use successful classification algorithms on mobile devices for

telemedicine purposes. Currently, few telemedicine products are available for analyzing heart sound recordings [11–14]. These telemedicine solutions take the heart sounds via a digital stethoscope and upload the recording to the cloud for classification.

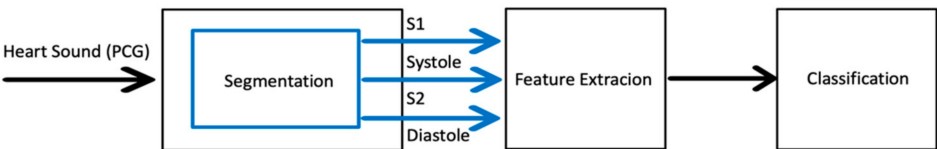

**Figure 1.** General steps for the automated heart sound classification tasks.

In this work, a mobile application designed to operate on mobile devices for telemedicine purposes is presented. The classification module of the software was trained and validated with the PhysioNet/CinC Challenge 2016 database. Since the PhysioNet dataset consists of abnormal recordings from coronary artery diseases, aortic stenosis, mitral regurgitation, mitral valve prolapse, and pathological murmurs conditions, the classification algorithm is expected to detect those abnormalities. The real-life success of the algorithm has been tested at the clinic using the mobile application. The algorithm can detect abnormal heart sounds in the validation set with 93% accuracy, and the mobile software that is powered by the same algorithm can detect abnormal samples with nearly the same sensitivity rate. The algorithm uses an ensemble nearest neighbors classifier, and features extracted from both Mel-frequency cepstral coefficients (MFFCs) and statistical properties of the fundamental heart sounds. Unlike available commercial products [11–13], the software works in an embedded fashion and does not need an Internet connection to upload the heart sound to the cloud for classification. Every five seconds, it analyzes the heart sound and prompts the predicted diagnosis to the screen.

The software presented in this work is intended to be used both in clinic settings by medical doctors and by ordinary people at home. Therefore, the software needs to be verified in real-life circumstances. For this, the software was installed on an Android device and the mobile application was tested at Ankara City Hospital, Cardiology Department by expert medical doctors. It is approved by the medical doctors that the artificial intelligence-powered mobile application can distinguish healthy and unhealthy individuals with 93% accuracy.

## 2. Materials and Methods

### 2.1. Data Set

For training and validation of the algorithm, the PhysioNet 2016 database which consists of 2430 labeled heart sounds was used. This dataset is a combination of nine heart sound databases collected by independent research groups at different times and with different devices. Since this database consists of nearly all of the available databases, it is widely used and accepted as the reference set for heart sound classification tasks [15–18]. All of the recordings are labeled as normal and abnormal, and the quality of the sounds, recording position, and patient profile are different. Since each database was evaluated by different medical staff and digital stethoscope devices, like all big datasets, the PhysioNet dataset is biased and includes personal judgment.

During the verification tests at Ankara City Hospital, the mobile application was used for 162 individuals. This verification dataset consists of 88 normal and 74 abnormal heart sound samples. All participants were between 15–70 years old and ThinkLabs One digital stethoscope was used by focusing multiple recording positions on the chest near the heart.

### 2.2. Software Description

Mobile software takes the stream data from a digital stethoscope. For every five seconds, the software extracts some features, then predicts and displays the result concurrently. Since the mobile application is designed to work in a connectionless fashion, all computations are made on-device. The workflow of the software is presented in Figure 2

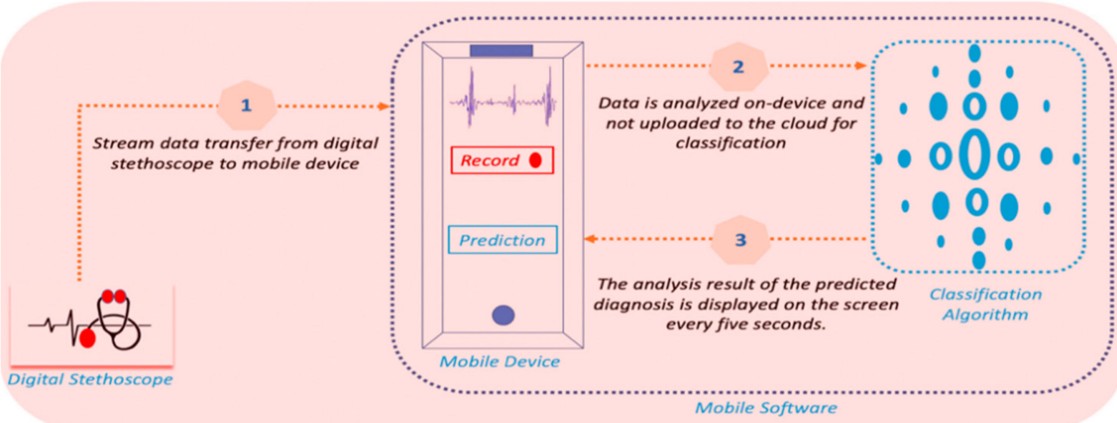

**Figure 2.** Workflow of the software. First, data is transferred from digital stethoscope to mobile device, then, the classification algorithm predicts the diagnosis, and the result is displayed simultaneously on the screen.

Software was developed in MATLAB by using the Simulink support package for mobile devices [19–22]. The support package has all the necessary blocks for using mobile devices' sensors. Moreover, it is possible to use MATLAB functions and the trained algorithm within the projects' block diagram. For using the trained algorithm, it should be converted to a C file beforehand, then it could be called back in the project. Generating C/C++ code from the trained algorithm can be done both by using the MATLAB Coder or by handcrafting the code [23].

Regarding the Simulink model presented in Figure 3:

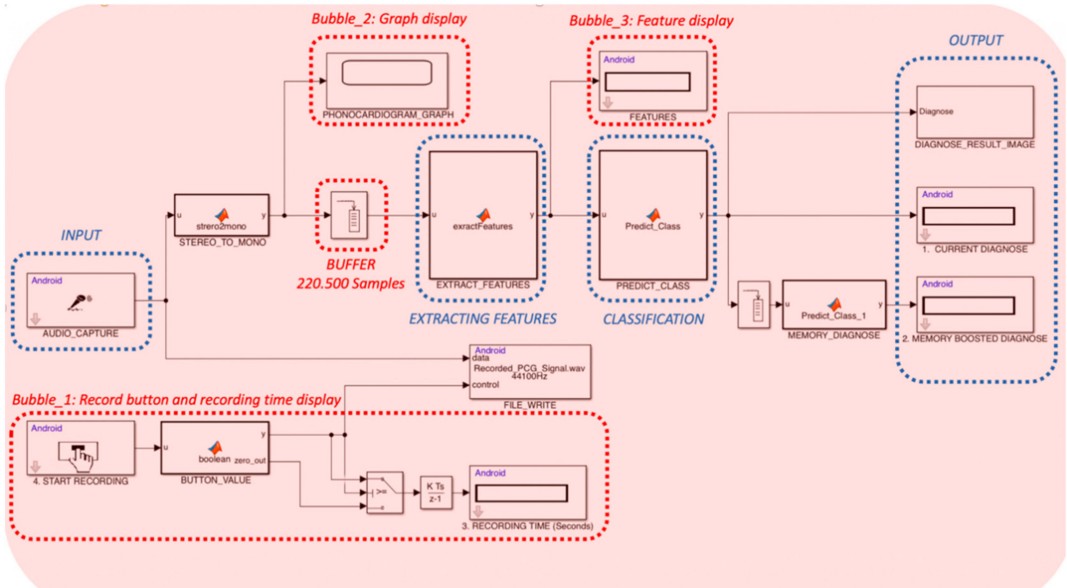

**Figure 3.** Simulink model of the software.

First, an audio capture block is used for getting the audio signal as input. This block enables to get a stereo audio stream from the microphone. In Bubble_1, there is a button for recording the stereo signal. If this button is switched on, recording time is displayed on the screen and the file write block writes the audio data to a wav type file on the mobile device. For further steps, stereo data are converted to a mono signal by a MATLAB function file. The converted signal is displayed in a graph using the Bubble_2 block.

Secondly, a buffer is used for accumulating the stream data. This buffer collects 220,500 samples from the stream source, the digital stethoscope in our case that has a sampling frequency of 44,100 samples per second. Then, the output of the buffer, a five

second-long signal, is transferred to the feature extraction block. Most of the features are acquired from Mel-frequency cepstral coefficients, which is a commonly used technique in speech recognition tasks [24,25]. This technique mimics the human ear's ability to scale the sounds by using a logarithmic scale function and a filter. The rest of the features are extracted from both statistical properties and the power spectrum of the signal. Those features are, respectively, standard deviation, kurtosis, spectral entropy, and maximum frequency. All extracted features are displayed on the screen (Bubble_3).

Third, extracted features are transferred to the classification bubble as input data for the classification algorithm. The classification algorithm predicts the label of five seconds long heart sound data whether it is a healthy or unhealthy sample. To determine the best classification algorithm, well-known methods were evaluated by using the PhysioNet database. During these experiments, PhysioNet data was divided into two parts, 70% for training and 30% for validation, and recordings used for training were not used for validation. According to the classification accuracy rate, the most successful algorithm is determined and used inside the *predict_Class* function block. The output of the *predict_Class* function block is a binary integer where the zero output value stands for healthy samples and the one output value stands for unhealthy samples.

The function of the output bubble in Figure 3 is to show the predicted diagnosis result on the screen as an integer value and to show the corresponding image of the result on the screen. From the input step to the output step, the software can process heart sounds around one second on a mobile device that has 1.2 GHz quad-core processor, 2 GB RAM capacity, and Android 4.4 operating system. This is a very satisfactory result because the software can prompt a diagnosis result of a five-second long instance after 1.1 s on a moderate mobile device.

The screen design of the mobile application is seen in Figure 4. On top of the right corner of the screen, the predicted diagnosis is shown by a binary value, and a corresponding diagnosis image is prompted on the left half of the screen for describing the algorithms' prediction by an image. When the predicted diagnosis image is slid to left, this part starts to show real-time phonocardiogram graphs. Below the predicted diagnosis display on the top right corner of the screen, recording time in seconds is displayed. Recording time is displayed according to the outputs of a counter function that is triggered by the boolean value of the *start recording* button located below the time display on the screen. When this button is switched on, the heart sound is also saved in the device's memory. At the bottom right of the screen, the extracted features are displayed every five seconds.

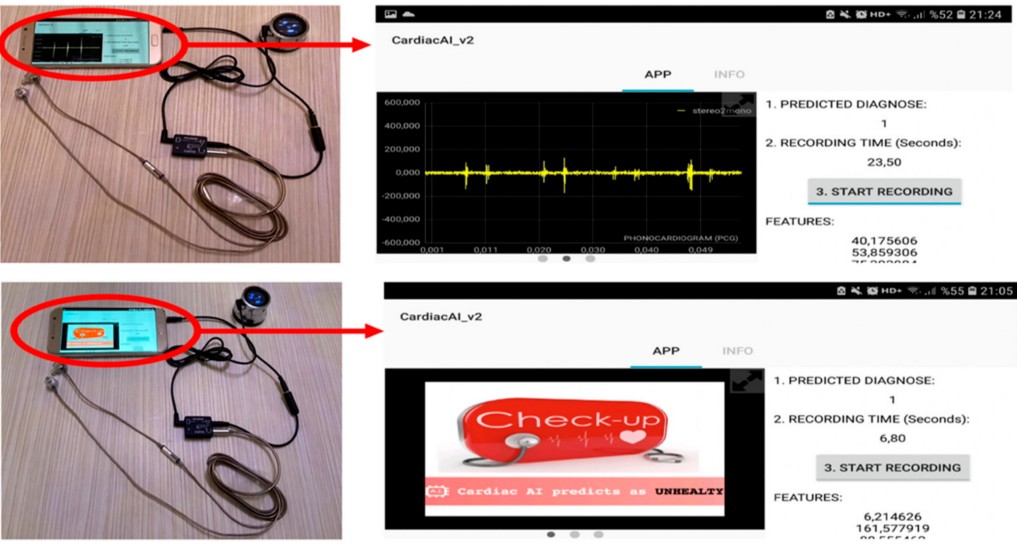

**Figure 4.** Mobile device and digital stethoscope connection on the left side. Screen design of the application (smartphone version) on the right side.

### 3. Results

From seven different machine learning techniques, a total of 25 different algorithms were tested with 15 features. PhysioNet dataset was separated into two, 70% for training and 30% for validation. All tests are conducted in MATLAB via the classification learner app. The detailed list of the tested algorithms is presented in Table 1.

**Table 1.** The algorithms used for tests.

| Index | Algorithms | | | | | | |
| --- | --- | --- | --- | --- | --- | --- | --- |
| | LR | NB | DA | DT | SVM | KNN | Ensemble |
| 1 | Logistic | Gaussian | Linear | Fine | Linear | Fine | Boosted Tree |
| 2 | | Kernel | Quadratic | Medium | Quadratic | Medium | Bagged Tree |
| 3 | | | | Coarse | Cubic | Coarse | Subspace DC |
| 4 | | | | | Fine | Cosine | Subspace KNN |
| 5 | | | | | Medium | Cubic | RUSBoosted Tree |
| 6 | | | | | Coarse | Weighted | |

LR = logistic regression, NB = naïve Bayes, DA = discriminant analysis, DT = decision tree, SVM = support vector machines, KNN = k-nearest neighbor.

Classification accuracy performance and detailed classification metrics of the tested algorithms are presented in Appendix A (Tables A1–A7). Among experimented algorithms, the most successful one is an ensemble model of which its learner type is nearest neighbors, the number of learners is 30, and the subspace dimension is eighth. The sensitivity of the ensemble model which shows the power of detecting diseases is 83.7% and the specificity of the model which shows the power of detecting healthy individuals is 96% (Figure 5). As seen from Figure 5, the algorithm can guess healthy individuals at a very high success rate but miss 16% of unhealthy samples.

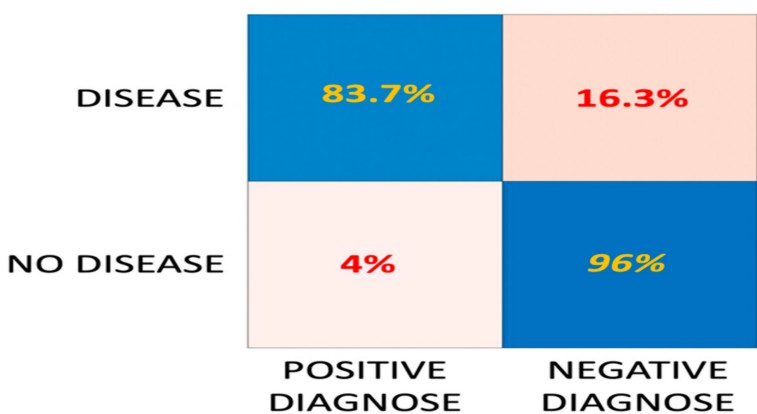

**Figure 5.** Detailed classification metrics of the ensemble model.

As stated in the previous part, the classification algorithm has been trained and validated with the publicly available PhysioNet 2016 database. For verification purposes, we used the mobile application in real-life circumstances at Ankara City Hospital. The cardiac experts tested the algorithm and mobile application over 162 individuals who visited the clinic for regular check-ups. All participants were labeled both by the cardiologists and by the mobile application. According to cardiologists, 88 people were detected as healthy and 74 people were detected as unhealthy. The unhealthy individuals determined by the cardiologists were forwarded to further diagnostic tests. In these tests, it was confirmed that all unhealthy diagnosed individuals have heart abnormalities. Therefore, the auscultation diagnoses of cardiologists were accepted as the ground truth data. Detailed information about the Ankara City Hospital recordings is presented in Table 2.

**Table 2.** Data set used in verification tests at Ankara City Hospital.

| Label | Auscultation Result | Confirmation Results | Length (Seconds) | Age (Years) | Recording Position | Sensor |
|---|---|---|---|---|---|---|
| Normal | 88 | - | 25–45 | 15–70 | Four Typical Position | ThinkLabs 44,100 Hz |
| Abnormal | 74 | 74 | 25–45 | 15–70 | Four Typical Position | ThinkLabs 44,100 Hz |

The recordings used in the verification phase are collected by the developed mobile application. Those recordings are of vital importance since they represent the data quality expected in clinical practice. Verification results that are presented in Table 3 show that the mobile application can detect normal heart sounds above 92% specificity and 81% sensitivity which is very close to validation results in Appendix A (Tables A1–A7). This result shows the robustness of the developed classification algorithm since it can work properly on a mobile device.

**Table 3.** Performance comparison of mobile application against validation/verification databases and medical staff.

| Performance Metric | Against Validation Data | Against Verification Data | Trained Cardiologists | General Practitioners |
|---|---|---|---|---|
| Specificity | 96.0% | 92.0% | 98.2% | 81.0% |
| Sensitivity | 83.7% | 81.0% | 69.6% | 31.0% |

In verification tests, an Android tablet was used and the prediction result of the application was determined by the memory boosted diagnosis display (Figure 6). The memory display is prompted by a function that accumulates the last five diagnoses in a buffer and tries to catch the four or more same results. Within 25 s, the application generates five diagnostic outputs, if four or five consecutive diagnoses are received, the memorized decision of the algorithm is determined, and this result is accepted as the final decision of the application.

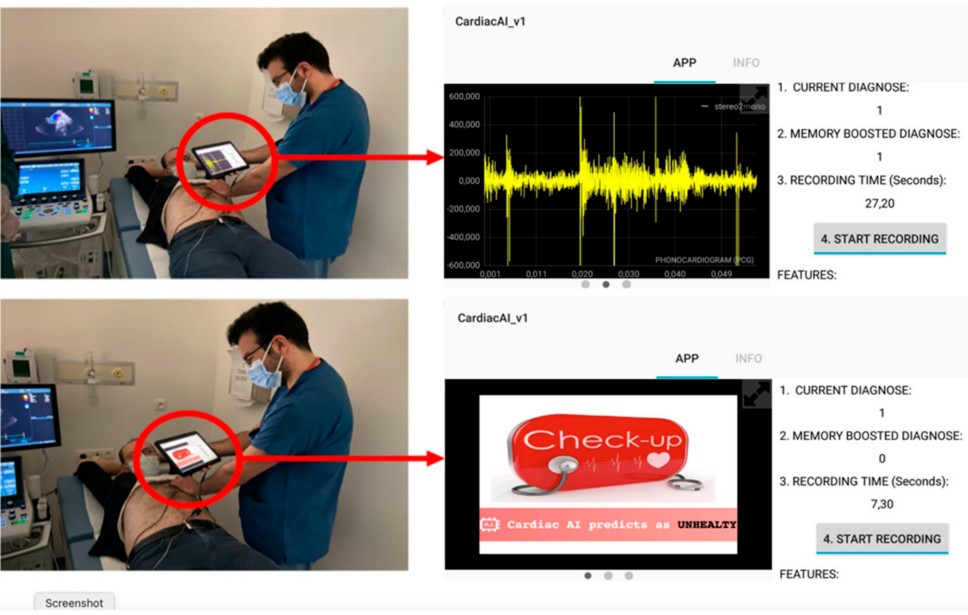

**Figure 6.** Verification tests in Ankara City Hospital and screenshot figures of the mobile application (tablet version).

## 4. Discussion

Medical doctors without special auscultation training cannot perform well in detecting murmurs. The accuracy of pediatricians in detecting heart murmurs is 24% [26]. Another study states that general practitioners can diagnose significant valvular heart diseases with a sensitivity of 43% and specificity of 69% but cardiologists can predict up to a sensitivity of 81% [27]. Even though some trained cardiologists can perform well, clinicians have variable auscultation performance [28,29]. Therefore, both cardiologists and non-cardiologist need a preliminary diagnostic tool to boost their auscultation accuracy. Similarly, non-trained clinicians have a low accuracy rate with electrocardiogram data and only cardiologists can diagnose successfully [30,31]. Moreover, electrocardiogram-based artificial intelligence models such as KardiaMobile and Eko AI software can perform as well as cardiologists [32]. Eko AI software requires users to upload their data to the cloud for computation. Unlike these commercial products, the developed application can run solely on the mobile device's hardware thanks to its informative features and robust classification algorithm. Regardless of having an internet connection, one can evaluate his or her heart health condition instantly with a sound recorder, preferably with a regular digital stethoscope. As a result, users do not need to upload the recordings to a server or a cloud for evaluation. This feature provides data privacy and flexibility for users. In the near future, with the proliferation of AI-powered applications for telemedicine purposes, data privacy issues will be more important.

Another key feature of the application is that it accepts the data from the start and it perpetually outputs the predicted diagnosis result in five-second periods. For future works, this property can be adopted to intensive-care units to trigger an alarm when an abnormality is detected. Cardiac event monitors control electrocardiogram signals to decide whether the patient's heart continues its function or not. Those patient monitoring systems do not provide any information about instant heart abnormalities. If this process can be converted to listening to the heart and continuously analyzing its condition, the effectiveness of patient monitoring systems will increase.

Given that we have a global pandemic, the medical system is under pressure, and clinicians are overwhelmed by the increasing number of patients caused by coronavirus disease 2019 (COVID-19). Telemedicine tools like the one presented in this study can reduce the number of healthy individuals visiting hospitals and as a result, take the burden off the medical system. Since the mobile application is designed to be used not only for experts but also by users of no medical education, verification data should consist of sounds that are recorded by ordinary users. To mimic the behavior of users with no medical background, during the verification tests, medical staff pointed the digital stethoscope at random locations on the chest near the heart. The verification results were successful but this is partly due to the digital stethoscope's filter property at the device level that can eliminate unwanted sounds produced from other organs such as lungs, etc. To compensate for the noise effect, an additional pre-processing block could be added in future works.

## 5. Conclusions

Without proper training, most clinicians fail to distinguish abnormal heart sounds from normal ones, but auscultation is one of the oldest, easiest, and main methods used for detecting heart diseases. To not miss a diagnosis, heart sounds should be listened to by artificial intelligence too. For this reason, we present a mobile application design that was developed in MATLAB. This application and its classification algorithm can work solely inside the device hardware and produces quick responses. In the verification phase, it was confirmed that the developed mobile application can work properly on mobile devices. As a result, the presented mobile application is proven to be a valuable diagnostic tool not only for clinicians but also for ordinary users.

**Author Contributions:** Conceptualization, M.G. and F.H.; methodology, M.G.; software, M.G.; validation, M.G. and F.H.; formal analysis, M.G.; investigation, K.Ö. and F.T.; resources, M.G., F.H. and K.Ö.; data curation, M.G., F.H. and K.Ö.; writing—original draft preparation, M.G.; writing—review and editing; M.G. and F.H.; visualization, M.G.; supervision, K.Ö. and F.H.; project administration, K.Ö. and F.H.; funding acquisition, M.G. and F.H. All authors have read and agreed to the published version of the manuscript.

**Funding:** This research received no external funding.

**Institutional Review Board Statement:** Not applicable.

**Informed Consent Statement:** Patient consent was waived due to routine auscultation being conducted on all patients, no invasive methods were used, and heart recordings of the participants were not saved in order to not infringe on their privacy. Verbal consent of all participants was collected beforehand for using the digital stethoscope with the AI-powered mobile application to compare diagnosis results of clinicians and algorithm.

**Data Availability Statement:** Android package files, codes, and feature set matrices are available online at https://github.com/mesuttguven/CardiacAI.git, and validation data set is available online at https://archive.physionet.org/pn3/challenge/2016/. Android app is available at the repository, everybody is encouraged to download, install, and test the mobile application to check the results presented here.

**Acknowledgments:** Authors would like to acknowledge the medical staff of the cardiovascular surgery department in Ankara City Hospital for their great effort and the TURK AI firm for donating the digital stethoscope and tablet devices.

**Conflicts of Interest:** The authors declare no conflict of interest.

## Appendix A

**Note:** Total number of observations is 13,015, and 70% of the observations were used for training, 30% were held for validation.

**Table A1.** Detailed classification performance of logistic regression classifiers.

| Algorithms | Accuracy | Sensitivity | Specificity |
|---|---|---|---|
| Logistic Regression Classifiers | 80.3% | 42.7% | 92.4% |

**Table A2.** Detailed classification performance of naïve Bayes classifiers.

| Algorithms | Accuracy | Sensitivity | Specificity |
|---|---|---|---|
| Gaussian Naïve Bayes Classifiers | 80.8% | 64% | 86.2% |
| Kernel Naïve Bayes Classifiers | 80.7% | 72.2% | 83.5% |

**Table A3.** Detailed classification performance of discriminant analysis classifiers.

| Algorithms | Accuracy | Sensitivity | Specificity |
|---|---|---|---|
| Linear Discriminant | 80.7% | 44.6% | 92.2% |
| Quadratic Discriminant | 82.1% | 48.8% | 92.8% |

**Table A4.** Detailed classification performance of discriminant analysis classifiers.

| Algorithms | Accuracy | Sensitivity | Specificity |
|---|---|---|---|
| Fine Tree | 86.6% | 68.3% | 92.4% |
| Medium Tree | 84.9% | 64.4% | 91.4% |
| Coarse Tree | 81.5% | 37.4% | 95.6% |

**Table A5.** Detailed classification performance of support vector machines (SVM) classifiers.

| Algorithms | Accuracy | Sensitivity | Specificity |
|---|---|---|---|
| Linear SVM | 80.7% | 43% | 92.7% |
| Quadratic SVM | 87.7% | 69.9% | 93.4% |
| Cubic SVM | 89.7% | 78% | 93.5% |
| Fine Gaussian SVM | 85.4% | 43.2% | 98.9% |
| Medium Gaussian SVM | 89.5% | 73.7% | 94.6% |
| Coarse Gaussian SVM | 85.6% | 58.6% | 94.2% |

**Table A6.** Detailed classification performance of k-nearest neighbor (KNN) classifiers.

| Algorithms | Accuracy | Sensitivity | Specificity |
|---|---|---|---|
| Fine KNN | 92.8% | 85.9% | 95% |
| Medium KNN | 90.9% | 82.7% | 93.5% |
| Coarse KNN | 87.1% | 64.5% | 94.3% |
| Cosine KNN | 89.2% | 75.7% | 93.5% |
| Cubic KNN | 90.4% | 81.8% | 93.1% |
| Weighted KNN | 92.1% | 83% | 95% |

**Table A7.** Detailed classification performance of ensemble classifiers.

| Algorithms | Accuracy | Sensitivity | Specificity |
|---|---|---|---|
| Boosted Trees | 87.7% | 68.7% | 93.8% |
| Bagged Trees | 90.9% | 79.5% | 94.5% |
| Subspace Discriminant | 76.3% | 18.1% | 94.9% |
| Subspace k-nearest neighbor | 93% | 84.6% | 95.6% |
| RUSBoosted Trees | 83.6% | 89.4% | 81.7% |

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
