# Peer review of "Heart Diseases Diagnose via Mobile Application"

_applsci, doi:10.3390/app11052430_

Round 1

Reviewer 1 Report

An interesting and well-planned study.

Author Response

Thank you very much for your time and interest in the study. We added and refined some parts in accordance with reviewers’ commands and the manuscript was examined the manuscript for the English language.

Reviewer 2 Report

In this manuscript, the authors investigated the use of artificial intelligence to diagnose heart murmurs via a mobile application. The application uses the PhisioNet 2016 database which is the largest public heart sound database and was developed in Mathlab. The data shows that this application can produce a quick and accurate response when compared with auscultation by experienced cardiologists.
The work is well written and the application of technology in medicine is certainly of great scientific interest, in fact in literature and on the market there are already devices with equal intentions

Comments:
- the authors should better specify the differentiation criteria among the various patients
- the authors should better argue the fact that heart murmurs have been discriminated clinically by different cardiologists, and therefore with methods of personal judgment, and with different stethoscopes
- were the heart murmurs diagnosed with the app then confirmed with imaging methods?
- the manuscript needs a revision of the English language

Author Response

(The authors gave the same response as above.)

Reviewer 3 Report

Introduction
The introduction should lead the reader to the topic. Thereby, the current state of science should be shown and gaps or open questions should be worked out. From these, the working hypothesis for the presented topic is then derived. This does not or only insufficiently take place in the presented article. Which abnormalities should be detected. 
Methodology
The description of the selection of an algorithm is usually a part of the results. The methodology should describe the criteria according to which the algorithm was selected. Also, the naming of the algorithms should give some information about the used configuration and the training performance. The computing time of the predictions is also interesting for operation on a mobile device and should be included in the consideration.
How was the criterion Accuracy formed in table one? Did you test your results for significance? How do you explain the high false positive rate of more than 16%?
Was this the same in the clinical test?

Results
The results are very brief and the data presented do not allow us to draw conclusions about the actual quality of the application. What was the false positive and false negative rate?
Not every congenital heart defect results in abnormal sounds that can be heard with a stethoscope. It is often the most complex heart defects where no abnormal heart murmur occurs at all. How should this be handled?

How was specificity and sensitivity calculated?

Discussion:
A classification of the result takes place in a few favorable literatures and an outlook is given. The small amount of data from subjects is regrettable. However, under the current circumstances not to change.

Another remark.
On the whole, the term artificial intelligence should be used with care, if mainly a small one-time statistical regression and / or optimization is in use. These are not intelligent, but conditioned and also not particularly new. Retrain does not take place and neither does the interaction of different algorithms. I recommend deleting Artificial Intelligence from the title. 

Author Response

Thank you very much for your time and interest in the study. We added and refined some parts in accordance with reviewers’ commands and the manuscript was examined the manuscript for the English language.

Point 1: Introduction; The introduction should lead the reader to the topic. Thereby, (a) the current state of science should be shown, and gaps or open questions should be worked out. From these, the working hypothesis for the presented topic is then derived. This does not or only insufficiently take place in the presented article. (b) Which abnormalities should be detected?  

Response 1:

(a) Current state of science, gaps and open questions emphasized by adding an extra part and a corresponding figure to introduction stated below.

  • The general methodology of the automated heart sound classification algorithms consist of three steps. Those are segmentation, feature extraction and classification steps respectively (Figure 1). Segmentation is necessary for classification since most of the features used for classification are derived from fundamental heart sounds (FHSs) that occur because of the contraction and relaxion movements of the heart [7]. Eventhough there are some other lower pithed sounds, FHSs usually include two distinct sounds, first (S1) and second (S2) heart sounds respectively. The time lapse between S1 and S2 is known as diastolic period and the time lapse between S2 and S1 is known as systolic period.
  • FIGURE 1 (added new)
  • After successfully segmenting the heart sounds, features are extracted from time, frequency, energy and high order statistics domain. Then, a suitable classification algorithm is determined. Various methods have been employed for classification such as k-nearest neighbors, artificial neural networks, support vector machines and hidden markov models [8-10]. It is possibble to use successful classification algorithms on mobile devices for telemedicine purposes. Currently, few telemedicine products are available for analyzing heart sound recordings [11-14]. These telemedicine solutions take the heart sounds via a digital stethoscope and upload the recording to the cloud for classification.

(b) The algorithm is expected to recognise abnormalities exist in the training dataset. This part is added to introduction as below.

-   Since PhysioNet dataset consist of abnormal recordings from coronary artery diseases, aortic stenosis, mitral regurgitation, mitral valve prolapse and pathological murmurs conditions, classification algorithm is expected to detect those abnormalities.  

Point 2: Methodology; (a) The description of the selection of an algorithm is usually a part of the results. (b) The methodology should describe the criteria according to which the algorithm was selected. (c) Also, the naming of the algorithms should give some information about the used configuration and the training performance. The computing time of the predictions is also interesting for operation on a mobile device and should be included in the consideration. (d) How was the criterion Accuracy formed in table one? Did you test your results for significance? How do you explain the high false positive rate of more than 16%? Was this the same in the clinical test?

Response 2:

(a) The parts which describe the selection of an algorithm (below part) were extracted from methodology and imported to results. “Table 1 and Figure 4” were extracted from methodology and imported to results section too.

-      Classification accuracy performance and detailed classification metrics of the tested algorithms is presented in Appendix A (Tables A1–A7). Among experimented algorithms, the most successful one is an ensemble model of which its learner type is nearest neighbors, number of learners is 30, and the subspace dimension is eighth. Sensitivity of the ensemble model which shows the power of detecting diseases is 83.7% and specificity of the model which shows the power of detecting healthy individual is 96% (Figure 5). As seen from the Figure 5, algorithm can guess healty individuals at a very high success rate but miss 16% of unhealthy samples

(b) Only the criteria used for the selection of classification algorithm is left in the methodology part. To emphasize the criteria, some additions were made (below part).

-      To determine the best classification algorithm, well-known methods were evaluated by using the PhysioNet database. During these experiments, PhysioNet data was divided into two parts, 70% for training and 30% for validation respectively. And recordings used for training were not used for validation.

(c) The computing time of the predictions is also interesting for operation on a mobile device and should be included in the consideration. For response the paragraph below is added.

- From the input step to output step, the software can process heart sounds around one second on a mobile device which has 1.2 Ghz. quad-core processor, 2 GB RAM capacity and Android 4.4 operating system. This is a very satisfactory result because software can prompt diagnose result of five-seconds long instance after 1.1 seconds on a moderate mobile device.

(d) How was the criterion Accuracy formed in table one? Did you test your results for significance?

      - Table 1 was replaced with a detailed appendix of tables which include accuracy, sensitivity and specificity metrics. Accuracy values was computed from the confusion matrix by “(True Positive Diagnose + True Negative Diagnose) / (True Positive Diagnose + True Negative Diagnose + False Positive + False Negative) “

       - Significance was taken into consideration in all tests. Training data is not equally distributed (do not have equal number of samples from each class), so this was compensated by using a misclassification error coefficient in training part. On the other hand, the confirmation dataset (Ankara City Hospital data) has equal samples (88 normal and 74 abnormal).

       How do you explain the high false positive rate of more than 16%? Was this the same in the clinical test?

- Even trained cardiologists, General practitioners have difficulties in detecting abnormal heart sounds. Please take look at the sensitivity rates (False Positive) from table 3 which is derived from references [30, 31].

Point 3: Results; (a) The results are very brief, and the data presented do not allow us to draw conclusions about the actual quality of the application. What was the false positive and false negative rate? (b) Not every congenital heart defect result in abnormal sounds that can be heard with a stethoscope. It is often the most complex heart defects where no abnormal heart murmur occurs at all. How should this be handled?

(c) How was specificity and sensitivity calculated?

(a) Results are brief, how to find FP and FN rate?

                       - Classification metrics (accuracy, sensitivity, specificity) of the tested algorithms were added by a new appendix. False positive rate can be calculated from sensitivity and false negative rate can be calculated from septicity rate in appendix tables.

(b) Not every congenital heart defect result in abnormal sounds that can be heard with a stethoscope. It is often the most complex heart defects where no abnormal heart murmur occurs at all.

                        - That’s a very good point. As far as I know, there are some papers claim to handle those recordings via deep learning. But in our case where all computations are done inside the mobile device hardware in a very moderate time (around one seconds), it is not suitable to implement a multi-layer neural network (deep learning). So, this is out of the scope of this study.

(c) How was specificity and sensitivity calculated?

                        -  Sensitivity rate = TP/ TP+FN and Specificity rate = TN/ TN+FP

Point 4: Discussion: A classification of the result takes place in a few favourable literatures and an outlook is given. The small amount of data from subjects is regrettable. However, under the current circumstances not to change.

                        - I think the amount of training and validation data sufficient enough since it is the combination of all heart sound records. On the other hand, the verification data, Ankara City Hospital (ACH) dataset used in verification tests are large enough to show the success of the algorithm and mobile software. The ACH may be larger than its current size but under COVID19 pandemic, the data has been formed in 6 mounts-long period and the medical doctors (me too) had COVID while this verification tests.

Point 5: Another remark. On the whole, the term artificial intelligence should be used with care, if mainly a small one-time statistical regression and / or optimization is in use. These are not intelligent, but conditioned and also not particularly new. Retrain does not take place and neither does the interaction of different algorithms. I recommend deleting Artificial Intelligence from the title.

                        - I deleted the “artificial intelligence” part from the title. The new title is “Heart Diseases Diagnose via Mobile Application “

Thank you for your valuable comments, I hope I would be able to address all the points you mentioned.

Round 2

Reviewer 2 Report

The authors responded comprehensively to the comments.
I have no further comments

Reviewer 3 Report

The authors have intensively dealt with the comments and suggestions and have fundamentally improved the submitted paper. I would like to thank the authors for this. I am looking forward to reading the published version.

This manuscript is a resubmission of an earlier submission. The following is a list of the peer review reports and author responses from that submission.